# Learning Programming Language in Higher Education for Sustainable Development: Point-Earning Bidding Method

**Tzer-Long Chen [1], Tsung-Chih Hsiao [2],\*, Tsan-Ching Kang [3], Ting-Yuan Wu [4] and Chih-Cheng Chen [5],[6],\***

1   Department of Finance, Providence University, Taichung 43301, Taiwan; tlchen1976@pu.edu.tw
2   School of Arts, Southeast University, Nanjing 211189, China
3   Department of Computer Science and Information Management, Providence University, Taichung 43301, Taiwan; tckang@pu.edu.tw
4   Department of Information Technology, Ling Tung University, Taichung 40852, Taiwan; tywu1995@gmail.com
5   School of Information Engineering, Jimei University, Xiamen 361021, China
6   Department of Aeronautical Engineering, Chaoyang University of Technology, Taichung 413310, Taiwan
\*   Correspondence: hsiaotc@gmail.com (T.-C.H.); 201761000018@jmu.edu.cn (C.-C.C.)

**Abstract:** Promoting learning passion for students in higher education is a vital and challenging issue. Due to the swift changes in society and technology development, many convenient and interesting technologies increasingly interfere with students' learning performances and reduce students' motivation for attending classes. For example, mobile games and social networking have led students to lose their passion for learning. MAPS (Mind Mapping, Asking Questions, Presentation, Scaffolding Instruction) is a flipped teaching model which has been proved as an effective approach to increasing students' reflective learning, allowing students to devote to self-learning and recovering their passion for learning. Thus, this study employed a MAPS teaching strategy that adopts a point-earning approach to encourage students to learn from peer feedback while the teacher can understand students' learning from the process. When the class is over, students can exchange the points for other rewards through a bidding mechanism which encourages students to regain passion for learning. The concept of Scaffolding Instruction points out that allowing students to sense positive peer pressure, which is caused by their peers' improvements, and the atmosphere where their peers are studying hard can enhance their learning motivation and reflection. The empirical results of this study found that peer assessment and feedback can improve the learning effectiveness of students with poor performance.

**Keywords:** education for sustainability; higher education; bidding mechanism; reflective learning; MAPS teaching model; peer assessment

---

## 1. Introduction

Enhancing students' enthusiasm for learning has always been a vital issue in higher education. With the advance of technology, however, many students are becoming increasingly addicted to mobile games and online communities—even in the classroom. These distractions affect their learning and mean they do not acquire the talents needed by industries upon their graduation. The phenomenon decreases students' capabilities after higher education. Therefore, it has become a challenge for teachers to find approaches to reclaim students' enthusiasm for learning. In response to environmental changes in higher education, researchers have proposed many innovative teaching methods. Studies have

found that the flip education model is one of the much-discussed approaches to recovering students' enthusiasm. Some well-known examples are: PaGamO developed by Professor Ping-Cheng Yeh, which adapted online games to stimulate students by rewarding points for answering questions and solving problems meaning that students gain the thrill of competition while learning. The other example is Cheng-Chung Wang who adopted the MAPS (Mind Mapping, Asking Questions, Presentation, Scaffolding Instruction) teaching model in teaching the Chinese language to increase students' language capacities through team competition and raise the capacity for peer learning and reflective learning, achieving teamwork learning effectiveness, and increasing self-learning and problem-solving skills so that students can maintain their interest in and enthusiasm for learning the programming language. This study explores new teaching methods designed to help students to revitalize their enthusiasm for learning and reflective learning.

Using the teaching concepts of MAPS, this study employed Scaffolding Instruction to implement a new teaching model that uses peer pressure and an independent learning environment. The model encourages students to share their work, engages them in after-class discussions and develops their problem-solving capabilities. In the course of guiding students, this study rewarded students with learning points through homework or quizzes. After class, a bidding mechanism was used to exchange learning points for rewards to stimulate greater learning enthusiasm. Peer assessment was also employed to stimulate the students of lower achievement. By analyzing students' grades, their learning progressed, and the effectiveness of the new teaching model was observed.

## 2. Literature Review

### 2.1. Flip Learning

Innovations in teaching practices often face obstacles, e.g., insufficient time for teaching innovation, parental rejection, and teachers' unwillingness to change. On the other hand, the success of teaching reform depends on learning effectiveness [1]. Allowing every student to learn independently is the foundation of higher education [2]. Flipped learning changes the relationship between teachers and students, allowing students to take on the role of active learning. It has been used in various teaching situations of higher education. Teachers provide students with the material for independent learning in higher education; however, they also need to keep track of student learning progress to enable effective learning, providing students with adequate rewards and encouragement at times. Competition and peer assessment can also serve as a way of rewarding learning. It can encourage students to collaborate and grow through constructive competition.

Teachers can shape a better learning environment to improve student performance and provide an opportunity for reflective learning. The most representative innovative flipped education method is PaGamO. PaGamO is learning software that uses an online battle game and assists students in effective learning in several ways, such as:

1. It can detect students' comprehension of basic concepts and allow teachers to fine-tune their subsequent teaching;

2. Students can do reflective learning, know their own learning status, and adjust their learning strategies accordingly, especially by learning to collaborate with others; and

3. The game questions are integrated with usual tests; thus, students who reach high scores in games also reach high scores in tests, which increases students' willingness to learn through the software [3].

### 2.2. Exploration in MAPS Teaching Method

The MAPS teaching method created by Wang set off an educational revolution, by advocating stimulating students' learning motivation through a daily "0.01 change" to flip their life [4,5]. A significant disparity in learning motivation is very common in rural schools and in higher education. Using the MAPS teaching method can solve the issue with collaborative learning. MAPS first classified

students into levels of A, B, C, and D based on their performance, and students of different levels were then mixed and regrouped. Students answered questions through teamwork. When the answer was made by a student of a higher level, lower points were awarded than when the answer was made by a student of a lower level. This differential reward system stimulated the learning motivation of low-level students and created a positive learning cycle for them [4]. MAPS consists of four parts: Mind Mapping, Asking Questions, Presentation, and Scaffolding Instruction. The function and purpose of each part are as follows:

(1) Mind Mapping: Help students to enhance their memory and expand the mind map of thinking.

(2) Asking Questions: Teachers design questions from various perspectives and help students to construct learning strategies through classroom questions and answers.

(3) Presentation: Through group presentations, teachers can determine if students learned, and if the system produced effective learning.

(4) Scaffolding Instruction. Due to the disparity in each individual student's starting point and motivation to learn, the heterogeneous group cooperative learning model is adopted. It is a useful method in higher education because students may have different background knowledge for a class. This allows low- and medium-level students to receive timely assistance from the high-level students in the group while further enhancing the reflective learning of high-achievement students [6,7].

### 2.3. QR Code

In 1994, the QR Code (Quick Response Code) was invented by a Japanese company called Denso-Wave. In 2000, it was recognized by the International Standard Organization (ISO) and became an international bar code standard. QR Codes can be scanned from any angle and present the stored information [8]. If the QR Code pattern has wear and cannot be recognized, it can still be read within an allowable blur range. Thus, the QR Code has high fault tolerance and is now a common technology used in everyday life [9].

The data storage capacity of the QR Code is 7089 numeric characters, 4296 alphanumeric characters, and 2953 binary characters. It uses the 984 characters of UTF-8 encoding and can represent roughly the amount of text contained in an A4 size paper. The QR Code is a square graphic as shown in Figure 1, using a 1:1:3:1:1 ratio of "black, white, black, white, and black" to enhance its interpretability. To improve the limitations of black-and-white graphic interpretation, various colors can now also be used in three distinctive squares at the corners of the QR Code to assist decoding software in direction interpretation [10]. Because QR Codes provide an easy way for students to read the stored information, they have been used frequently by teachers in higher education in recent years.

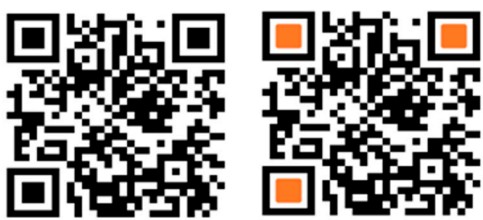

**Figure 1.** QR Code (Quick Response Code) matrix barcode technology.

### 2.4. Information Education and Learning Assessment Development

For students taking an Information major, the focus of programming courses lies not just in memorizing program syntax but also in learning to use programming logic to solve problems. It is the main teaching goal for students to learn a programming language in higher education. However, current programming interfaces are mostly instructional editing interfaces instead of graphic interfaces which could raise the learning performances and increase the students' interest in learning programming [1,11]. Additionally, in the process of program writing, it is difficult for teachers to immediately understand the students' learning conditions and the difficulties they are encountering. This is a challenge for

programming learning in higher education [7]. To understand students' progress, evaluating an individual student's work is the most common practice currently. However, using assessment results to facilitate students' learning outcomes is a different and complex issue [12–16]. Peer assessment may be one of the solutions. In past literature, it was found that peer assessment has a positive effect on students' reflective learning, and it motivates students to do extracurricular activities to get higher scores [17,18]. Incorporating peer assessment in teaching also provided a better learning environment [19,20].

### 2.5. R Language

R Language is an operating environment for writing programs and was developed by Ross Ihaka and Robert Gentleman in 1993. The primary use of R Language is in data analysis. It can not only find the relationship between data (e.g., data mining) but also provide users with powerful graphic functions for data visualization. R Language can be executed on a variety of platforms, using a text interface or a visually graphic interface (e.g., Rstudio). R Language can easily expand its functions through kits to meet the needs of various users [21,22]. This study employed R Language to produce a relationship chart to observe the relationship between student achievement and teacher achievement.

## 3. System Architecture and Implementation

### 3.1. System Architecture

This study proposed combining MAPS of educational concepts and APP (Application) development to enhance students' interest in learning. To provide teachers with an innovative new teaching tool, the study employed low-cost and popular QR Codes as a vehicle in building an innovative teaching model with an answer points and bidding system. The system architecture is shown in Figure 2. Adopting cloud application, the study provided students and teachers, through the Internet service, with the software and hardware developed based on the actual needs to stimulate students' enthusiasm for learning and students' behaviors for reflective learning.

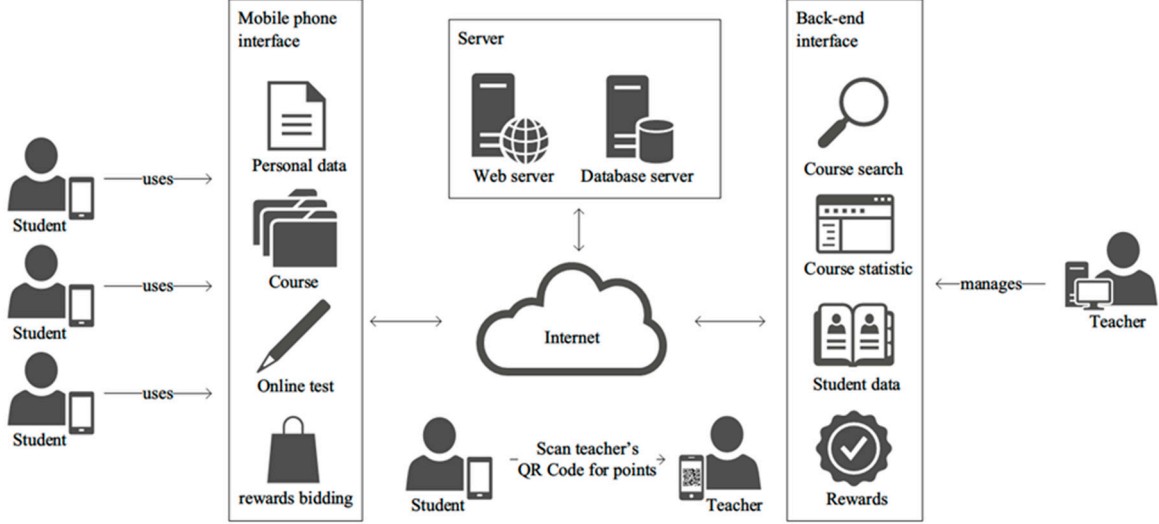

**Figure 2.** System architecture.

There are three parts in this system architecture: Student Interface, Teacher Interface, and Management Interface. The functions of the Student Interface include:

(1)  taking course exams,
(2)  collecting learning points;
(3)  bidding with learning points to get rewards;

(4)     reviewing learning achievement statistics; and

(5)     conducting account management.

The functions of the Teacher Interface include:

(1) uploading tests, producing a QR Code for the test (students acquire the test by scanning the QR Code and receive learning points after passing the test);

(2) querying and modifying class and student statistics; (3) uploading rewards data; (4) ruling on bids for popular rewards; and

(5) conducting account management.

The functions of the Management Interface include:

(1)     managing the Teacher and Student Interface accounts;

(2)     maintaining the system and webpage servers; and

(3)     managing online tests and collecting point data.

## 3.2. System Implementations

The software and hardware used for the system developed in this study are shown in Table 1.

**Table 1.** Development environment.

| | |
|---|---|
| Software | • Android Studio<br>• Eclipse<br>• Apache HTTP Server<br>• PHP 7<br>• MariaDB 10<br>• QR Code<br>• Microsoft Windows 10 Pro<br>• CentOS 7<br>• Rstudio<br>• Scratch |
| Hardware | • Wireless Access Points<br>• Desktop and laptop computers<br>• Android mobile phones |

## 3.3. Database Planning

This study employed MySQL's branch database MariaDB, which supports standard SQL (Structured Query Language) syntax, APIs (Application Program Interface) and command columns. Based on the system's functional requirements, the data table and the fields of the database are planned as follows:

1.     User: The fields of the table include Uid, Email, Uname, Upasswd, Point, and Uprimess.
2.     Exam: The fields of the table include Ectopic, Eoption_1, Eruption _2, Eoption_3, Eoption_4, and Answer.
3.     Shop: The fields of the table include Sid, Sitem, Sprint, and Picture.
4.     History: The fields of the table include Uid, Eid, and Epoint.
5.     History: The fields of the table include Uid and Sid.

The association between the data sets in this research database is shown in Figure 3.

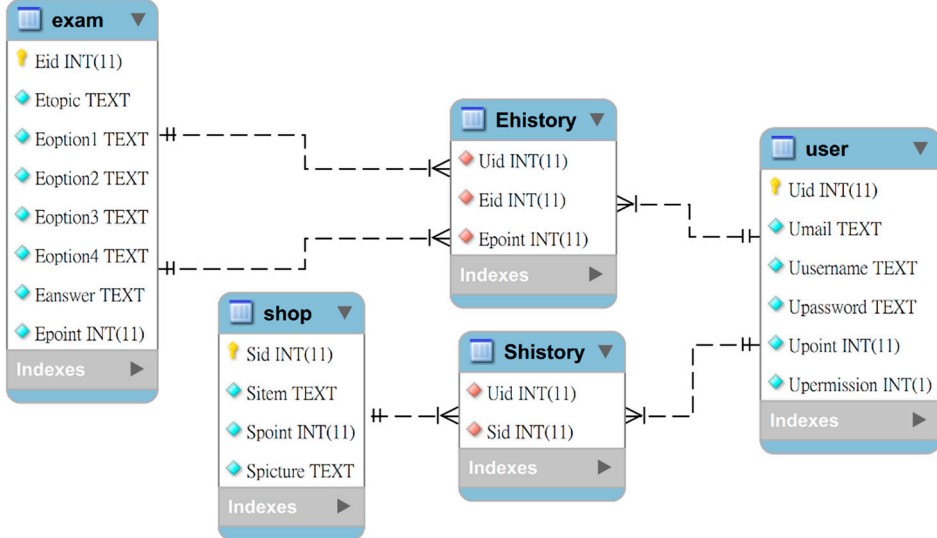

**Figure 3.** Database planning.

### 3.4. Learning Point System Processes

Students can take a test after installing the app. By passing the test, they earn learning points. In addition, teachers can use the Hash Function to generate a QR Code for one-time learning points as an incentive for students to actively participate. Students redeem points to do competitive bidding for prizes. Students first enter the learning points they want to bid for a prize. If the points are above a minimum amount of points that is used to exchange rewards, and exceed bids by other students, they can successfully redeem the prize. The process of earning and redeeming learning points is shown in Figure 4.

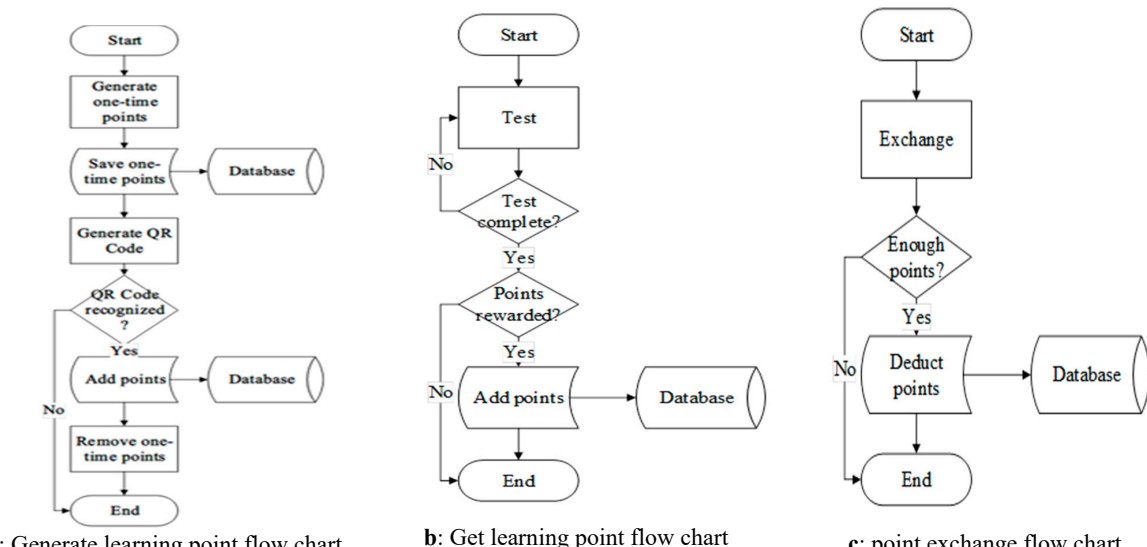

**a**: Generate learning point flow chart     **b**: Get learning point flow chart     **c**: point exchange flow chart

**Figure 4.** The learning point system.

### 3.5. Mobile Phone Interface Designs

The mobile phone interface design includes the Student Interface, the Teacher Interface, and the Management Interface. After students log into the system, they can choose "My Area", "Instant test", "Reward Bidding", "Scan QR Code" or "Setting". In "My Area", students can view course-related information. In "Instant test", students can receive tests from teachers. In "Reward Bidding", students can redeem prizes by bidding more learning points than other students. Lastly, when teachers want to

encourage the student to focus on the class, teachers can generate a QR Code and allow students to scan the QR Code to obtain learning points. The mobile phone interface design on the Student Interface is shown in Figure 5. The functions of the Teacher and Management Interfaces include uploading test items, gathering statistics of student data, uploading pictures of reward items, reviewing the numerical range of test results, and exporting it into a grade report.

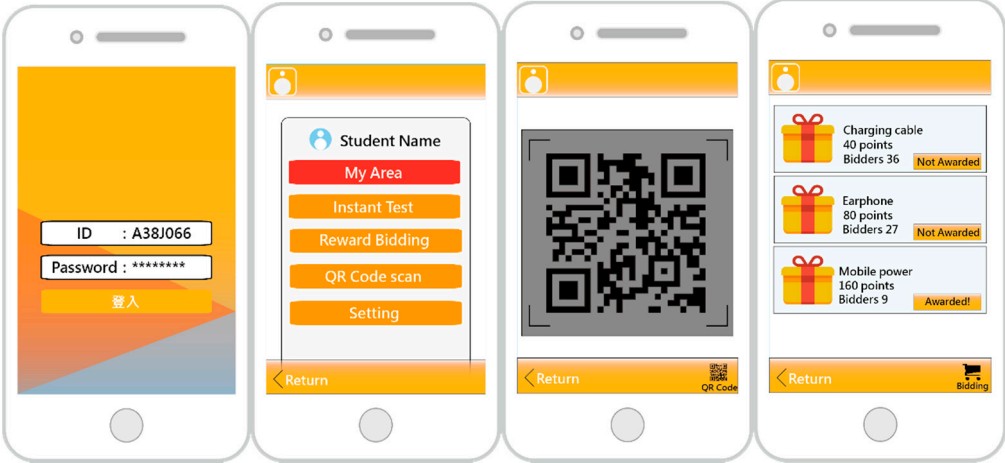

**Figure 5.** Student End mobile phone interface design.

### 3.6. Course Designs and After-Class Assessment Analysis

The purpose of this study is to allow students to learn a programming language efficiently. The programming language students learned in this study is Scratch, which was developed by the Massachusetts Institute of Technology (MIT) in 2007. Scratch can operate on a browser to write and execute programs. Users can develop programs without installing special software so it provides a good environment for beginners. Scratch uses a visual programming interface, which is more easily accepted by students than the text interface used by many other programs. Students only need to drag the blocks to complete a simple program, which significantly lowers the threshold of programming knowledge required.

The criteria for peer assessment are shown in Table 2; Table 3. Table 2 lists the scoring criteria for each score from 1 to 5 points. Each score has a clear description of the criteria. To prevent students from scoring errors due to insufficient understanding of the scoring criteria, before conducting peer assessment, the teachers must give a detailed explanation of the scoring criteria. Table 3 lists the program language learning objectives that each score is scheduled to achieve; listed respectively are the processing logic, component usage, program simplification, appearance, and functionality.

Teachers can understand students' professional level through analyzing the gap between students and their scores; they can also understand whether the current course content meets the teaching objectives. Through analyzing professional approaching status, students can understand whether their learning meets the teacher's expectation. The score sheet is shown in Table 4.

$$\text{Students' professional level} = \frac{\text{Average score of peer assessment}}{\text{Score of teacher assessment}} \tag{1}$$

**Table 2.** Assessment criteria (Holistic).

| Score | Levels | Descriptions on Criteria |
|---|---|---|
| 5 | Excellent | Can independently think of a set of methods for program logic processing and add additional components to design layout and function; the code is concise and contains expansion functions; the appearance is specially set through external components; the implementation results and display standards. |
| 4 | Very Good | Can roughly think through the problem; uses predetermined components to set functions; the code is concise and can complete the functions required; the preset items are adjusted and modified for the appearance performance and present the correct program functions. |
| 3 | Good | Uses other's processing logic to implement the code; uses too many components to set the functions; the code is not simplified; the appearance design is fair; the functionality is roughly complete and occasional errors occur. |
| 2 | Fair | Needs other's assistance to complete, with some difficulty, the processing method; uses fewer components to set the functions; the appearance design is not fully arranged; the function is roughly normal. |
| 1 | Insufficient | Cannot complete coding and overall appearance to the requirement of the test; cannot understand the problem and cannot deconstruct the process; lacks component usage; and the code is too long to execute. |

**Table 3.** Criteria and learning goals (Analytic).

| | Excellent | Very Good | Good | Fair | Insufficient |
|---|---|---|---|---|---|
| Processing logic | Develops a processing method independently. | Follows thinking pattern of others but implements independently. | Follows processing method of others. | Needs others' assistance to implement. | Cannot implement to the question. |
| Component usage | Additional components are added and functional. | The components function to expectations. | Uses excessive components that do not affect functions. | Components used are not functional. | Pieces together components inadequately. |
| Program simplification | The code is concise with additional functions. | The code is streamlined and achieves implementation goals. | The code is not streamlined but achieves implementation goals. | The code functions adequately and can perform setting. | The code file is lengthy and doesn't meet the implementation goals. |
| Appearance | Uses extra appearance components for typesetting. | The appearance design and typeset is good. | The appearance design is adequate but could use polishing. | Only a small part of appearance is arrayed properly. | The appearance design is not well-done. |
| Functionality | Fully functional and includes expansion functions. | The functions perform correctly and produce results. | The functions are generally normal with occasional unexpected results. | The function is generally normal with frequent unexpected results. | Does not function normally. |

**Table 4.** Student professional approach table.

| Numerical Range | Description |
|---|---|
| >1+ | Students and teachers share the same comprehension approach. |
| >0.95 | Have a general understanding of the teaching objectives. |
| >0.8 | Have a slight difference in scores. |
| >0.75 | Can still mutually understand the way of scoring. |
| >0.6 | There are differences in mutual scoring approach. |
| <0.6 | Major intellectual gap between students and teachers. |

After each peer assessment, students can calculate their own growth. Their value level and improvement over time lets them understand their learning progress. The detailed score sheet is shown in Table 5.

$$\text{Students' self} - \text{growth change} = 1 - \frac{n^{th} \text{ assignment}}{(n-1)^{th} \text{ assignment}} \tag{2}$$

**Table 5.** Student self-growth changes table.

| Numerical Range | Changes in Learning Status |
|---|---|
| >0.4 | Significant progress from benchmark value |
| >0.2 | Significant progress difference |
| 0.0 | Room for improvement |
| <−0.2 | Moderate decline in progress |
| <−0.4 | Significant decline from benchmark value |

Based on the results of peer assessment, teachers reward students with learning points according to the reward tables (as shown in Table 6). The number of learning points rewarded are 5 points for scores of 90 or above, 3 points for scores of 80–89, 2 points for scores of 70–79, 1 point for scores of 60–69, and 0 points for scores of less than 60. Students can view from the app on their mobile phones the exchangeable items and the learning points required to redeem the items.

**Table 6.** Learning points table.

| Score | Learning Points Rewarded |
|---|---|
| ≥90 | 5 |
| 80–89 | 3 |
| 70–79 | 2 |
| 60–69 | 1 |
| <60 | 0 |

Table 7 shows the items for redemption. Students can accumulate the learning points earned for the whole semester to exchange for higher-end prizes. In addition to earning learning points from the scores of peer assessment, students may also acquire points from accomplishing tasks assigned by the teacher or answering questions in class.

<div align="center">**Table 7.** List of prizes.</div>

| Learning Points Required | Prizes |
|:---:|:---:|
| 10 points | Tetra Pak drinks |
| 20 points | Canned drinks |
| 40 points | USB charging cable |
| 80 points | Earphones |
| 160 points | Mobile power pack |

## 4. Learning Effectiveness Analysis

A total of 37 college students participated in the study. The participants were divided into 12 groups in a mobile phone programming course. The details of peer assessment and the teacher's professional assessment are shown in Figure 6. The student peer assessment and teacher score results are shown in color-coded point diagrams, marking the group numbers and group scores received, respectively. The differences displayed in Figure 6 were used to observe whether student peer assessment results aligned with the teacher's professional score criteria or demonstrated significant disparity. Figure 6 shows the distribution of various student groups and teacher scores.

Figure 7 shows the differences in students' and teachers' assessments at a professional level. Students are represented by dots, and the teachers are represented by triangles. The vertical axis of the chart represents the scores of peer assessment and teacher assessment, respectively. The horizontal axis of each chart represents students' professional approaching degree. It can be observed that in most groups, the assessment results of students and teachers were close, and the score of students was often lower than the score of teachers.

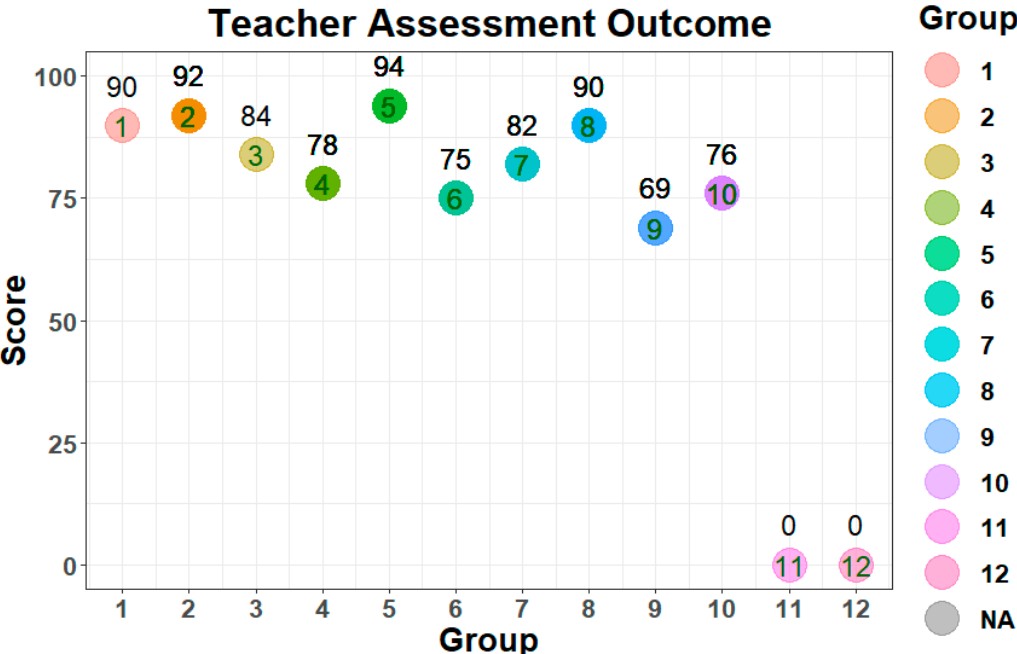

**a**: Teacher assessment outcomes

**Figure 6.** *Cont.*

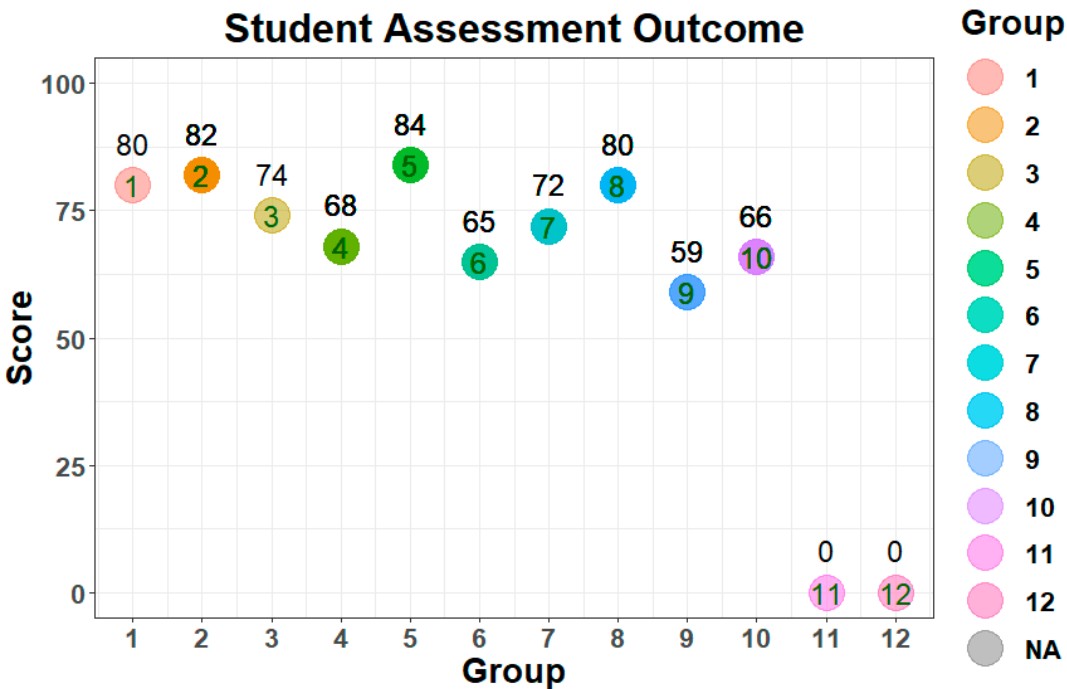

**b**: Students assessment outcomes.

**Figure 6.** Students and teacher assessment outcomes.

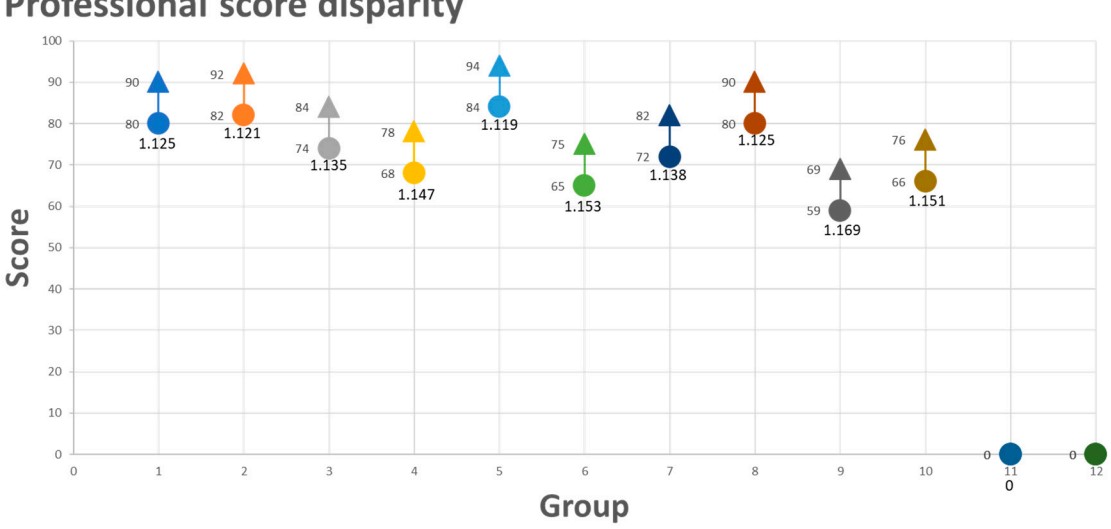

**Figure 7.** Professional score disparity. Notes: Dot: students; Triangle: teacher.

Figure 8 shows the learning status of the respective students. In the figure, a dot represents the scores of the first assignment and a triangle represents the scores of the second assignment. The vertical axis represents the score value achieved for the assignment, and the horizontal axis represents the value change of students' professional self-growth. It can be observed that most students made progress in the second assignment; however, it also shows that three students showed significant declines (<−0.4).

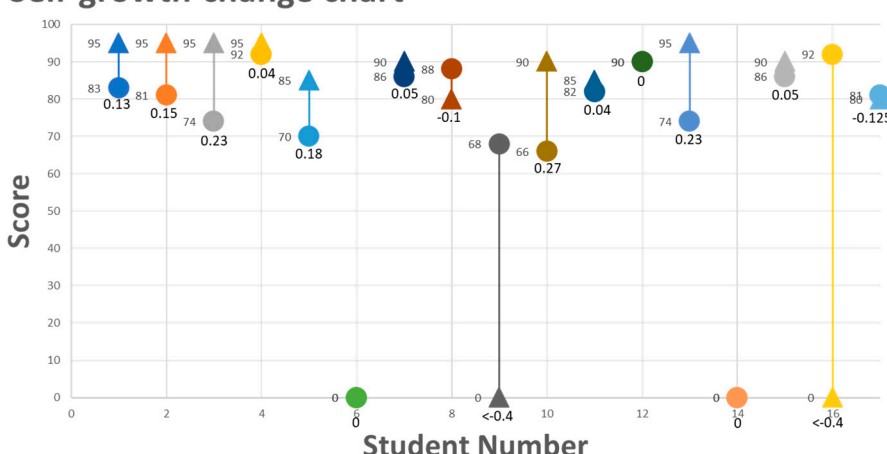

**a**: 1st assignment Self-growth change charts

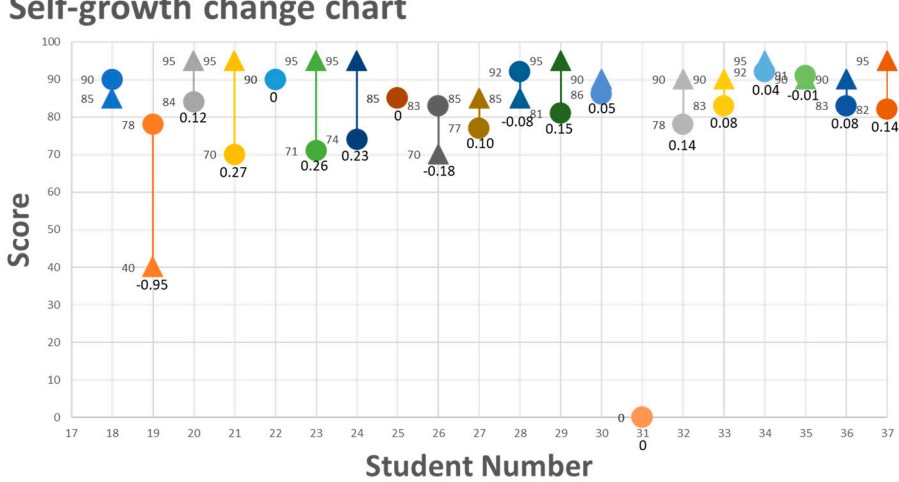

**b**: 2nd assignment Self-growth change charts

**Figure 8.** Self-growth change charts. Notes: Dot: 1st assignment; Triangle: 2nd assignment.

## 5. Conclusions

Based on MAPS theory, this study developed a software application that can enhance students' course participation. A teacher can teach students with the software following the four steps of MAPS theory, which are Mind Mapping, Asking Questions, Presentation, and Scaffolding Instruction. Before the class began, the teacher announced to students the problem to be solved and divided students into groups for separate discussions on solutions. Then, students used the software to conduct peer assessment among themselves to raise students' learning motivation. With the software, the teacher can flip the traditional learning approach and enhance students' learning effectiveness.

The software has become one of the effective communication channels between the teachers and students. The functions of the software include homework assignment, online tests, and point-exchanging for items. Students can operate the software through mobile phones and their learning records can be stored in the cloud; teachers and students can perform learning effectiveness analysis through these learning records. The software collects both peer assessment and teacher assessment to calculate students' performance. Through peer assessment, students can learn what

other students think about their work, which can generate competitive pressure for improvement. The software can assist students and teachers in after-class analysis to have a better understanding of personal learning effectiveness. The software also designed in an incentive mechanism to enhance students' learning motivation. Students can earn learning points by passing the tests and exchange the points earned for prizes. The results of this study confirm that the software can assist students in independent learning and guide students to regain their enthusiasm for learning through integrating information technology and innovative teaching.

**Author Contributions:** Conceptualization, T.-L.C. and T.-C.H.; Methodology, T.-L.C.; Software development, T.-Y.W.; Formal analysis, T.-C.K. and C.-C.C.; Writing—Original draft preparation, T.-L.C.; Writing—Review and editing, T.-L.C. and T.-C.K.; Funding acquisition, T.-L.C. and T.-C.H. All authors have read and agreed to the published version of the manuscript.

**Funding:** This research was funded by Projects of Natural Science Foundation of Fujian Province of China (Nos. 2017J01109), Science and Technology Planning Fund of Quanzhou (Nos. 2016T009) and the MOE Teaching Practice Research Program (PEE107142).

**Acknowledgments:** This work was supported by the Natural Science Foundation of Fujian Province of China (Nos. 2017J01109), Science and Technology Planning Fund of Quanzhou (Nos. 2016T009) and the MOE Teaching Practice Research Program (PEE107142).

**Conflicts of Interest:** The authors declare no conflict of interest.

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
