# Peer review of "Learning Programming Language in Higher Education for Sustainable Development: Point-Earning Bidding Method"

_sustainability, doi:10.3390/su12114489_

Round 1
Reviewer 1 Report
The research is very interesting and the theme is of most importance in the current context.
The research is very well developed and scientific writing is of top-level.
I suggest to improve the conclusions, make it more specific and linked to the literature review.
Author Response
Response to the comments of reviewers:
Reviewer 1:
Comments and Suggestions for Authors
The research is very interesting and the theme is of most importance in the current context. The research is very well developed and scientific writing is of top-level. I suggest to improve the conclusions, make it more specific and linked to the literature review.
Answer: Thank you for your comments.
Based on MAPS theory, this study developed a software application that can enhance students’ course participation. A teacher can teach students with the software following four steps of MAPS theory, which are Mind Mapping, Asking Questions, Presentation and Scaffolding Instruction. Before the class begins, the teacher announced to students the problem to be solved and divided students into groups for separate discussions for solutions. Then students used the software to conduct peer assessment among students to raise students’ learning motivation. With the software, the teacher can flip the traditional learning approach and enhance students’ learning effectiveness.
The software becomes one of the effective communication channels between teachers and students. The functions of this software include homework assignment, online tests, and point-exchanging for items. Students can operate through mobile phones and their learning records can be stored in the cloud; teachers and students can perform learning effectiveness analysis through these learning records. The software collects both peer assessment and teacher assessment to calculate students’ performance. Through peer assessment, students can learn what other students think about their work, which can generate competitive pressure for improvement. The software can assist students and teacher in after-class analysis to have a better understanding of personal learning effectiveness. The software also designed in an incentive mechanism to enhance students’ learning motivation. Students earn learning points by passing the tests and exchange for prizes with the points earned. The results of this study confirmed that the software can assist students in independent learning and guide students to regain their enthusiasm for learning through integrating information technology and innovative teaching.
We included the above discussions in the revised manuscript (page 13).

Reviewer 2 Report
Dear authors
hoping that you are well, in these difficult times, I present my comments.
More than ever, we have to continue producing quality research, which contributes to the improvement of learning, increasingly in virtual settings. It is a commitment to future generations that must continue to learn more and more online, taking advantage of the potential available. Thus, the research that is carried out and published must describe effective learning scenarios that promote motivation, collaboration and meaningful learning in stimulating virtual learning contexts.
Your investigation has disclosed an experience of this type, with great transformative potential. That is why I congratulate you and I believe that it must be internationalized so that this know-how can be replicated in other contexts.
It is rigorously documented, based on credible research principles and with sufficient evidence of the methodological process that led to the described credible results.
My congratulations. Keep safe and in good health.
Author Response
Response to the comments of reviewers:
Reviewer 2:
Comments and Suggestions for Authors
hoping that you are well, in these difficult times, I present my comments. More than ever, we have to continue producing quality research, which contributes to the improvement of learning, increasingly in virtual settings. It is a commitment to future generations that must continue to learn more and more online, taking advantage of the potential available. Thus, the research that is carried out and published must describe effective learning scenarios that promote motivation, collaboration and meaningful learning in stimulating virtual learning contexts.
Your investigation has disclosed an experience of this type, with great transformative potential. That is why I congratulate you and I believe that it must be internationalized so that this know-how can be replicated in other contexts. It is rigorously documented, based on credible research principles and with sufficient evidence of the methodological process that led to the described credible results. My congratulations. Keep safe and in good health.
Answer: We are glad that you see this study as being an interesting and important one for the education community. We very much appreciate your positive and constructive feedback on our manuscript.
